# HDAC3 Activity is Essential for Human Leukemic Cell Growth and the Expression of β-catenin, MYC, and WT1

**DOI:** 10.3390/cancers11101436

**Published:** 2019-09-26

**Authors:** Mandy Beyer, Annette Romanski, Al-Hassan M. Mustafa, Miriam Pons, Iris Büchler, Anja Vogel, Andrea Pautz, Andreas Sellmer, Günter Schneider, Gesine Bug, Oliver H. Krämer

**Affiliations:** 1Department of Toxicology, University Medical Center, 55131 Mainz, Germany; manbeyer@uni-mainz.de (M.B.); alabdeen@uni-mainz.de (A.-H.M.M.); miripons@uni-mainz.de (M.P.); 2Department of Medicine II, Hematology/Oncology, University Hospital, 60590 Frankfurt, Germany; a.romanski@blutspende.de (A.R.); iris.buechler@gmail.com (I.B.); a.vogel@blutspende.de (A.V.); g.bug@em.uni-frankfurt.de (G.B.); 3Department of Pharmacology, University Medical Center, 55131 Mainz, Germany; pautz@uni-mainz.de; 4Institute of Pharmacy, Department of Pharmaceutical/Medicinal Chemistry I, University of Regensburg, 93040 Regensburg, Germany; andreas.sellmer@chemie.uni-regensburg.de; 5Klinik und Poliklinik für Innere Medizin II, Technical University of Munich, 81675 Munich, Germany; guenter.schneider@tum.de

**Keywords:** AML, β-catenin, HDAC, HDACi, indomethacin, molecular marker, MYC, WT1

## Abstract

Therapy of acute myeloid leukemia (AML) is unsatisfactory. Histone deacetylase inhibitors (HDACi) are active against leukemic cells in vitro and in vivo. Clinical data suggest further testing of such epigenetic drugs and to identify mechanisms and markers for their efficacy. Primary and permanent AML cells were screened for viability, replication stress/DNA damage, and regrowth capacities after single exposures to the clinically used pan-HDACi panobinostat (LBH589), the class I HDACi entinostat/romidepsin (MS-275/FK228), the HDAC3 inhibitor RGFP966, the HDAC6 inhibitor marbostat-100, the non-steroidal anti-inflammatory drug (NSAID) indomethacin, and the replication stress inducer hydroxyurea (HU). Immunoblotting was used to test if HDACi modulate the leukemia-associated transcription factors β-catenin, Wilms tumor (WT1), and myelocytomatosis oncogene (MYC). RNAi was used to delineate how these factors interact. We show that LBH589, MS-275, FK228, RGFP966, and HU induce apoptosis, replication stress/DNA damage, and apoptotic fragmentation of β-catenin. Indomethacin destabilizes β-catenin and potentiates anti-proliferative effects of HDACi. HDACi attenuate WT1 and MYC caspase-dependently and -independently. Genetic experiments reveal a cross-regulation between MYC and WT1 and a regulation of β-catenin by WT1. In conclusion, reduced levels of β-catenin, MYC, and WT1 are molecular markers for the efficacy of HDACi. HDAC3 inhibition induces apoptosis and disrupts tumor-associated protein expression.

## 1. Introduction

It is estimated that there will be 437,033 new cases of leukemia and 309,006 deaths associated worldwide in 2018 [1]. Predictions state that more than 21,450 people will be diagnosed with acute myeloid leukemia (AML) and nearly 10,920 people will die from it alone in the USA in 2019 [2]. This warrants the search for new therapeutic strategies. The transcription factor β-catenin appears as an actionable drug target in AML, because the expression and activity of β-catenin is linked to disease initiation, unfavorable karyotypes, and poor prognosis. Cells from such patients show enhanced self-renewal capacity, which suggests that β-catenin contributes to a stemness phenotype [3,4,5,6,7,8,9]. Moreover, β-catenin is expressed at significantly higher levels in AML compared to acute lymphoblastic leukemia and β-catenin was found to be overexpressed in 16/25 and 13/59 primary AML samples [5,6]. Furthermore, a recent report illustrates that β-catenin is important for chemotherapy-associated senescence and contributes to aggressive growth patterns of leukemia and lymphoma cells [10]. However, patient-to-patient variations regarding the addiction of AML cells to β-catenin [4,5,11] demonstrate the need to further characterize how β-catenin affects leukemogenesis.

Secreted factors of the wingless-type MMTV integration site (WNT) family stabilize β-catenin. They reduce the phosphorylation of β-catenin and its subsequent poly-ubiquitylation for proteasomal degradation. Consequently, β-catenin accumulates in the nucleus, where it induces genes in complex with the transcription factor, T cell factor-4 [12,13]. In AML cells, a cell-intrinsic activation of β-catenin can render leukemic stem cells independent of niche-derived WNT signals [14]. Factors that are responsible for the proteasomal degradation of β-catenin include the glycogen synthase kinase-3 (GSK3), the adapter adenomatous polyposis coli, E3 ubiquitin ligases such as seven-in-absentia-homologues-1/-2, F-box/WD repeat-containing protein 1A, and other proteins [15]. In addition to this proteasomal pathway, β-catenin can be cleaved by caspases [16,17]. 

Current research strives to identify and exploit epigenetic mechanisms for the therapy of solid tumors and leukemia. A dysregulation of histone deacetylases (HDACs) is seen in leukemic cells and four Histone deacetylase inhibitors (HDACi; LBH589, FK228, vorinostat (SAHA), and belinostat (PXD101)) have been approved by the FDA for the treatment of hematological disorders. Moreover, there are currently at least 15 HDACi in clinical trials [18,19]. The development of specific HDACi aims to define and exploit individual functions of HDACs in vivo and to reduce adverse effects of HDACi treatment [18,20,21]. Clinically used HDACi encompass small molecules that inhibit all zinc-dependent HDACs or HDAC subclasses ([18,19] and http://www.clinicaltrials.org). For example, the hydroxamic acid LBH589 blocks zinc-dependent HDACs that belong to class I (HDAC1, -2, -3, and -8), class II (HDAC4, -5, -6, -7, -9, and -10), and class IV (HDAC11). The benzamide MS-275 and the depsipeptide FK228 act only against the class I HDACs HDAC1, -2, and -3 [22].

A recent report shows that the anthraquinone oxime-analog BC2059 attenuates the expression of β-catenin in AML cells with and without an internal tandem duplication of the leukemogenic kinase FMS-like tyrosine kinase (FLT3-ITD) through a proteasomal mechanism. This mechanism involves a displacement of the scaffold protein transducin β-like-1 from β-catenin [23]. This work also indicates that LBH589 and BC2059 combine favorably against AML cells and that this effect is associated with a reduction of β-catenin [23]. However, it has not been addressed how HDACi decrease β-catenin, whether the attenuation of β-catenin by HDACi is a key factor for the fate of AML cells, and if one of the 11 zinc-dependent mammalian HDACs controls β-catenin expression. One of these HDACs might be the class IIB deacetylase HDAC6, because the tubulin polymerization promoting protein-1 blocks HDAC6 and attenuates β-catenin expression [24]. Furthermore, HDAC6 interacts with the cancer stem cell marker membrane glycoprotein CD133 and both were reported to stabilize β-catenin acetylation-dependently [25]. However, a key impact of HDAC6 on an important developmental regulator like β-catenin [12,13] is hard to reconcile with the lack of phenotypic abnormalities in HDAC6 knockout mice and with accumulating evidence that an inhibition of HDAC6 does not affect cancer cell growth [26,27,28,29]. HDACi that preferentially target HDAC3 evoke a destabilization of β-catenin, but not its mRNA expression, in breast cancer cells [30]. Therefore, HDAC3 could be a key positive regulator of β-catenin stability in leukemic cells.

There is evidence that non-steroidal anti-inflammatory drugs (NSAIDs), which inhibit the immunomodulatory enzymes cyclooxygenase-1/cyclooxygenase-2 (COX1/COX2), antagonize β-catenin expression and the stemness of leukemic cells [31,32,33,34]. Moreover, data collected with cells from various solid tumors suggest that a combined application of indomethacin with pan-HDACi as single or bifunctional molecules suppresses cell proliferation and angiogenesis. Remarkably, normal cells are spared by such treatment [35,36]. Whether NSAIDs combine favorably with HDACi against leukemic cells is an interesting option that remains to be investigated.

The transcription factor myelocytomatosis oncogene (MYC), which has been implicated in maintaining the stemness of AML cells [13,37,38], can be a target gene of β-catenin in leukemic cells [23,39,40]. Pan- and class I HDACi provoke a downregulation of MYC in cancer cells [19,23,41,42,43,44]. In addition to β-catenin and MYC, the transcription factor Wilms tumor (WT1) is associated with cell differentiation and leukemogenesis [45], and, like β-catenin and MYC, WT1 is regulated by HDACi [46]. Furthermore, WT1 can regulate the expression of MYC and of β-catenin positively and negatively [47,48,49,50,51,52,53,54]. It is currently not clear if there is a mutual regulation of these factors in leukemic cells and whether HDACi alter a potential interplay between these three transcription factors.

In order to fully develop the therapeutic potential of HDACi, some key questions need to be resolved. For example, 18 human HDACs fall into the classes I, II, III, and IV and it has to be further clarified if a pharmacological inhibition of subsets or even single HDACs can produce anti-leukemic effects. Moreover, further molecular mechanisms and markers have to be identified to use HDACi effectively. We set out to answer these questions. Our data demonstrate that HDAC3 activity is a key factor for the survival and the maintenance of tumor- and stemness-associated transcription factors in AML cells.

## 2. Results

### 2.1. Responses of Leukemic Cells to LBH589

We analyzed the responses of human AML cell lines (FLT3-ITD-positive MV4-11 and JAK2V617F-positive HEL cells) to the pan-HDACi LBH589. We noted a significant time- and dose-dependent accumulation of annexin-V-FITC-positive cells. After 24 h, 10 nM LBH589 induced a three-fold increase in annexin-V-FITC-positive MV4-11 cells, which corresponds to 16% apoptotic cells. 30 nM LBH589 caused a four-fold increase of apoptosis in MV4-11 cells and a six-fold increase of apoptotic cells in HEL cells, which corresponds to 34% and 67% dead cells (Figure 1A; see Appendix A for exemplary dot plots).

After 48 h, 10 nM LBH589 sufficed to induce 78% annexin-V-positive cells in MV4-11 cell cultures and 30 nM LBH589 led to 78% annexin-V-positive cells in HEL cell cultures (Appendix A).

The analysis of the cell cycle distributions of LBH589-treated cells revealed that 10 nM LBH589 significantly increased the number of MV4-11 cells in the G1 phase by 20% and reduced the number of S phase cells by 18%. Such changes also occurred in HEL cells as a trend. 30 nM LBH589 augmented the subG1 fraction, which represents cells with fragmented DNA, by 23% in MV4-11 and by 30% in HEL cell cultures (Figure 1B). These increased levels of the subG1 fractions were linked to a decline of the G1 and S phase populations in both cell lines and 30 nM LBH589 reduced the number of HEL cells in S phase significantly by 9% (Figure 1B).

In both cell types, the novel and specific HDAC6 inhibitor marbostat-100 [26] caused a slight and insignificant increase in their G1 phase populations at the expense of S phase populations. Marbostat-100 did not induce apoptosis in MV4-11 and HEL cell cultures (Figure 1A,B and Appendix A). Thus, pro-apoptotic effects of LBH589 are unlikely caused by its inhibitory effect on HDAC6.

To corroborate these results, we analyzed further apoptosis markers, the cleavage of the executioner caspase-3 and the caspase-dependent cleavage of PARP1 [29,55]. Congruent with our flow cytometry analyses, we detected significant caspase-3 activation and cleaved PARP1 in MV4-11 and HEL cells incubated with 30 nM LBH589 for 24 h (Figure 1C). Compared to HEL cells, MV4-11 cells have much lower levels of PARP1, which are more evident as cleaved PARP1 in apoptotic MV4-11 cells. It is currently unknown whether HDACi induce PARP1 that is subsequently cleaved or if our antibodies recognize cleaved PARP1 better than its full-length form. 

We additionally analyzed the ability of these cells to regrow at low density once the treatment with HDACi had been stopped. We treated MV4-11 and HEL cells with 10 or 30 nM LBH589 for 24 h. Thereafter, the cells were harvested and washed twice for withdrawal of HDACi. The proliferation of the cells was determined at day 4 after discontinuation of the treatment. Regrowth of MV4-11 was nearly prevented by 10 nM LBH589 and HEL cells were unable to regrow after an exposure to 30 nM LBH589 (Figure 1D).

These data indicate that low levels of LBH589 mainly promote cell cycle arrest and higher doses lead to the induction of apoptosis and the fragmentation of DNA in MV4-11 and HEL cell cultures.

### 2.2. Reduction of β-Catenin upon Class I HDAC Inhibition

Since β-catenin is relevant for the growth and survival of leukemic cells [3,4,5,6,7,8,9], we tested whether LBH589 alters the expression of this protein in leukemic cells. Western blot analyses of MV4-11 and HEL cells revealed that β-catenin was significantly degraded after incubation with LBH589. This led to the advent of an additional band that was revealed by the β-catenin antibody (Figure 2A). The β-catenin mRNA levels were not decreased by LBH589 (data not shown).

As LBH589 blocks class I, II, and IV HDACs [22], we addressed whether class I HDACs and/or HDAC6 prevent the processing of β-catenin. We inhibited HDAC1-3 with MS-275 [22] and HDAC6 with marbostat-100 [26]. We noticed that MS-275 caused the cleavage of PARP1 and β-catenin together with an activation of caspase-3 in both cell lines (Figure 2A). LBH589 and MS-275 also promoted an accumulation of ɣH2AX indicating replication stress and/or DNA damage [21] in MV4-11 and HEL cells (Figure 2A). We could confirm [26] that LBH589 attenuates the expression of HDAC6 in leukemic cells. Moreover, LBH589 as well as marbostat-100 effectively inhibit HDAC6, which is evident as accumulation of acetylated tubulin [26,28,29]. Congruent with Figure 1A,B, marbostat-100 did not evoke the activation of caspase-3 or the processing of β-catenin (Figure 2A). 

To exclude that the processing of β-catenin in response to HDACi is an artifact of permanent leukemic cell cultures, we incubated the LTC FFM12 with LBH589. As in permanent cell lines, LBH589 led to a cleavage of β-catenin and PARP1 (Figure 2B).

Next, we compared the β-catenin levels of MV4-11 and HEL cells with the β-catenin levels in normal PBMCs. We included colorectal cancer cells with wild-type (RKO cells) or mutant β-catenin (HCT116 cells) for comparison. We found a similar expression of β-catenin in PBMCs, RKO, HEL, and MV4-11 cells. HCT116 cells showed the expected [56] overexpression of mutant β-catenin (Figure 2C). These data suggest that β-catenin is wild-type in MV4-11 and HEL cells.

We noted the occurrence of β-catenin cleavage products in cells with a high activity of the apoptosis executioner enzyme caspase-3 (Figure 2A), which can cleave β-catenin [16,17]. Therefore, we speculated that caspases process β-catenin in AML cells exposed to pro-apoptotic doses of LBH589. We tested this hypothesis with the pan-caspase inhibitor z-VAD-FMK. Incubation of AML cells with LBH589 and z-VAD-FMK prevented the occurrence of β-catenin cleavage products and restored the levels of full-length β-catenin (Figure 2D). We observed similar data for MS-275 and FK228 and the efficiency of z-VAD-FMK was proven as abrogated activation of caspase-3 and intact PARP1 (Appendix A).

To complement these analyses, we tested for putative alterations of the expression and phosphorylation-dependent activation of GSK3β, which promotes the proteasomal degradation of β-catenin [15]. We noted that HDACi do not alter total and phosphorylated GSK3β in MV4-11 and FFM12 cells (Figure 2B,D).

LBH589, MS-275, and FK228 increased the levels of ɣH2AX and promoted apoptosis of MV4-11 cells (Figure 2A, and Appendix A). DNA double strand breaks, which are generated upon DNA fragmentation by caspase-activated DNase, can also induce ɣH2AX [57,58]. Therefore, such an accumulation of ɣH2AX might be a consequence of apoptotic DNA fragmentation. To test this, we incubated MV4-11 cells with HDACi and z-VAD-FMK. We found that this accumulation of ɣH2AX could only be partially abrogated with z-VAD-FMK (Appendix A). These data agree with previous reports indicating that HDACs are gatekeepers of genomic integrity [21,28,59,60,61].

We summarize that LBH589 induced apoptosis concomitant with an attenuation of β-catenin.

### 2.3. Indomethacin Can Accentuate Anti-Proliferative Effects of LBH589

During the analysis of various AML cell lines, we noticed that the human myoblast cell line KG1 was sensitive to apoptosis induction by LBH589, but retained β-catenin (Figure 3A–C). We observed a dose-dependent accumulation of acetylated histones and of acetylated tubulin in LBH589-treated KG1 cells (Figure 3A,B). This was associated with a dose-dependent cleavage of PARP1 (Figure 3A,C) and a significant increase in early and late apoptotic cells in KG1 cell cultures (Appendix A). Such an independence of apoptosis induction and β-catenin processing was also found in the AML-LTC FFM05 (data not shown).

Differences in the HDACi-mediated regulation of β-catenin in MV4-11 cells and KG1 cells cannot be explained by differences in total and phosphorylated GSK3β, as these are unaltered by HDACi (Figure 2D and Figure 3C). Moreover, we could prove the efficacy of LBH589 in both MV4-11 and KG1 cells as accumulation of acetylated forms of histone H3 and tubulin (Figure 3A,B).

A common response of primary and permanent cell lines to LBH589 is a reduction of MYC (Figure 3B,C and Appendix A) and this occurs irrespective of β-catenin in KG1 cells (Figure 3B,C). Furthermore, in contrast to the caspase-dependent processing of β-catenin in MV4-11 cells (Figure 2D), the HDACi-mediated attenuation of MYC cannot be rescued by anti-apoptotic concentrations of z-VAD-FMK in KG1 cells (Figure 3C).

The frequently prescribed NSAID indomethacin [62] has been reported to reduce β-catenin expression in chronic myeloid leukemia (CML) cells [33]. We noted a detectable effect of indomethacin against β-catenin in KG1 cells (Figure 3D). Therefore, we tested whether a reduction of β-catenin by indomethacin combines favorably with LBH589 against KG1 cells. KG1 cells hardly recovered at low density from a previous treatment with 30 nM LBH589 until day 4 after treatment discontinuation. A near complete inhibition of proliferation until day 4 was caused by 50 nM LBH589. Indomethacin dose-dependently suppressed the recovery of cells after drug removal and potentiated long-lasting anti-proliferative effects of LBH589 at a low dose of 10 nM (Figure 3E).

These data suggest that indomethacin promotes LBH589-evoked anti-leukemic effects.

### 2.4. HDAC3 Activity Is Required to Maintain β-Catenin, MYC, and WT1

Specific HDACi are considered as agents that can eliminate cancer cells at a low cost of side-effects [18,19,20]. Since HDAC3 is required for the survival and genomic integrity of leukemic cells [59,60,63,64,65], we speculated that the pro-apoptotic and DNA-damaging effects of LBH589 and MS-275 are due to an inhibition of HDAC3. To test this, we blocked HDAC3 with its specific inhibitor RGFP966 [59,60,63,65,66]. We applied a maximal dose of 10 µM because RGFP966 does not inhibit any other HDAC at a dose up to 15 µM in a biochemical assay [66]. Flow cytometry of fixed, propidium iodide (PI)-stained cells MV4-11 cells revealed that treatment with RGFP966 for 24 h dose-dependently triggered an increase of the subG1 fractions in AML cell cultures. 1, 5, and 10 µM RGFP966 increased the subG1 fractions from 8% to 11%, 18%, and 25%, respectively. This effect was significant for 5–10 µM RGFP966 and it was associated with a significant reduction of cells in S phase and G2/M phase (Figure 4A).

Pan-caspase inhibition with z-VAD-FMK prevented DNA fragmentation measured as subG1 fraction and the reduction of cells in G2/M phase by 10 µM RGFP966. Moreover, z-VAD-FMK in combination with 5–10 µM RGFP966 increased the numbers of cells in G1 phase significantly. The reduction of S phase cells by RGFP966 was though not affected by z-VAD-FMK (Figure 4A). These findings suggest that RGFP966 causes apoptosis of MV4-11 cells and that this prevents an accumulation of cells in the G1 and G2/M phases in response to RGFP966.

Annexin-V staining verified that 1, 5, and 10 µM RGFP966 increase apoptosis from 4.6% to 5.4%, 16%, and 25%, respectively in MV4-11 cell cultures. This experiment also shows that 5–10 µM RGFP966 evoke a significant increase in cells in late apoptosis (Figure 4B; see Appendix A for exemplary dot blots). This notion is consistent with the typical occurrence of fragmented DNA during the late apoptotic process (Figure 4A). Consistent herewith, we observed an apoptotic cleavage of caspase-3 and PARP1 (Figure 4C).

It is often unknown if the inhibition of one single HDAC is sufficient to attenuate oncogenic proteins. Therefore, we tested for a potential impact of RGFP966 on the leukemia- and stemness-associated transcription factors β-catenin, MYC, and WT1. We found that RGFP966 reduced WT1 and led to the advent of a WT1 cleavage product (Figure 4C), which is seen in leukemic cells undergoing apoptosis [29,55]. A loss of WT1 has consistently been noted in K562 cells treated with FK228 [29]. RGFP966 also attenuated the expression of β-catenin and MYC in a dose-dependent manner and led to the occurrence of a β-catenin cleavage product (Figure 4D). 

As expected [59,60,63], RGFP966 also caused an accumulation of ɣH2AX indicating replication stress and DNA damage (Figure 4D). To assess if this increase in ɣH2AX is linked to DNA damage or a consequence of DNA fragmentation by apoptosis [57,58], we incubated MV4-11 cells with 1 µM RGFP966. We chose this dose as it does not evoke significant DNA fragmentation (Figure 4A) and stained cells for 53BP1 foci by immunofluorescence. We chose this marker by virtue of its ability to stain DNA breaks within foci [67]. We could readily detect 53BP1 foci formation associated with 53BP1 nuclear translocation in RGFP966-treated MV4-11 cells (Figure 4E), indicating DNA damage.

Due to the accumulation of ɣH2AX and 53BP1 foci in response to HDACi (Figure 2A, Figure 4D,E, and Appendix A), we asked whether the reduced stability of β-catenin, MYC, and WT1 could also occur upon replication stress/DNA damage. Therefore, we assessed a putative regulation of these proteins by hydroxyurea (HU). We chose this clinically widely used drug, because it is a bona fide inducer of replication stress and DNA damage. Moreover, HU leads to a caspase-dependent processing of WT1 in leukemic cells [29,55]. HU dose-dependently reduced full-length β-catenin and led to the appearance of cleaved β-catenin. Furthermore, HU reduced MYC and caused an accumulation of ɣH2AX without altering the acetylation levels of histone H3 (Appendix A).

A loss of proliferation-associated transcription factors as well as replication stress/DNA damage can cause a permanent growth arrest [13,21]. Assessment of regrowth capacities showed that RGFP966 doses that significantly evoked apoptosis also restrained proliferation after drug removal. Moreover, 5 µM RGFP966 were about as effective as 30 nM LBH589 (Figure 4F).

These data suggest that HDAC3 is necessary for AML cell survival and the maintenance of β-catenin, MYC, WT1, and the prevention of replication stress and DNA damage.

### 2.5. Interplay between β-Catenin, MYC, and WT1

Since there is ample evidence for an interaction of β-catenin, MYC, and WT1 at multiple levels (e.g., [13,23,37,38,39,40,45,68]), we tested with RNAi whether these factors regulate each other in MV4-11 cells.

On the one hand, a reduction of MYC led to a reduction of WT1, indicating that MYC promotes the expression of WT1. On the other hand, a reduction of WT1 increased the levels of MYC (Figure 5A,B). Furthermore, a reduction of WT1 increased the levels of β-catenin, signifying that WT1 negatively regulates both β-catenin and MYC. A knockdown of β-catenin produced no effects on MYC and WT1 (Figure 5A,B). Flow cytometry assessing PI showed that the reduction of these three proteins up to 48 h did not affect the vitality of MV4-11 cells (Figure 5B). Hence, the interplay between MYC and WT1 is not a consequence of cell death.

These data illustrate that MYC and WT1 regulate each other and that WT1 reduces β-catenin levels in MV4-11 cells.

## 3. Discussion

We showed that LBH589 evoked growth arrest and apoptosis in leukemic cells. As this effect was also seen with MS-275, FK228 and RGFP966, we concluded that class I HDACs, and particularly HDAC3, were necessary for the survival of leukemic cells. The compromised survival of HDACi-exposed leukemic cells is associated with an accumulation of the replication stress/DNA damage marker ɣH2AX and with the formation of 53BP1 foci. 1 µM RGFP966 evokes replication stress/DNA damage without massive apoptotic DNA damage and z-VAD-FMK cannot abrogate ɣH2AX accumulation in HDACi-treated leukemic cells. From these data we deduced that HDACi-induced replication stress/DNA damage precedes apoptotic DNA fragmentation, which can also trigger the formation of ɣH2AX [57,58]. Others also report that a pharmacological interference with HDAC3 triggers replication stress, DNA damage, cell death, and enhanced chemosensitivity of lymphocytic and myeloid leukemia cells [59,60,63,65,69]. Such data are consistent with the reported impaired DNA repair capacities of HDACi-treated leukemic cells [21,28,59,60,61,63,64,70].

As HDAC3 has functions that are dependent and independent of its catalytic activity [71] it is important that RGFP966 phenocopies a genetic depletion of HDAC3 in transformed hematopoietic cells [65]. Since up to 15 µM RGFP966 specifically inhibits HDAC3 [66], our experimental settings with 1–10 µM RGFP966 likely revealed specific functions of HDAC3 activity. Hence, targeting HDAC3 seems to be a promising strategy to decrease β-catenin, MYC, and WT1, and to eliminate leukemic cells through replication stress/DNA damage and apoptosis. Our flow cytometry data demonstrated that RGFP966 decreased the numbers of cells in the G1 and G2/M phases dependent on caspases. The S phase populations were not rescued though when z-VAD-FMK was applied with RGFP966. This also applied to the accumulation of ɣH2AX, which was largely unaffected by z-VAD-FMK. Therefore, we concluded that S phase stalling and replication stress/DNA damage induction by RGFP966 was due to a genuine induction of replication stress/DNA damage that halted cell cycle progression. Further studies are underway to determine the targets of HDAC3 that control cellular responses to replication stress and DNA damage and whether cells slip from G2/M phase to G1 phase or die out of G2/M phase through mitotic catastrophe.

We further show that the extent of β-catenin processing by caspases tied in with a failure of leukemic cells to recover from HDACi-induced stress. Accordingly, a destabilization of β-catenin with indomethacin combined superiorly with HDACi. This finding could be due to the destabilization of β-catenin by both HDACi and NSAIDs, but NSAIDs exert multiple effects [35,62,72,73]. However, while several NSAIDs repress transcription factors of the NF-κB family, which can be a driver of tumorigenesis in hematopoietic and solid tumor cells, indomethacin does not inactivate NF-κB, AP1, and cancer-relevant kinases. Nonetheless, indomethacin activates PPARɣ [74] and an activation of PPARɣ by glitazones can attack the leukemic stem cell pool in CML [75]. Additional investigations are justified to define the beneficial effects of NSAIDs on leukemic cells and their stem cells. This also applies to solid tumor cells form breast, colon, lung, and prostate, for which bifunctional conjugates of indomethacin with SAHA are potent inhibitors of growth and angiogenesis [35,36]. For example, compound 11b, an indomethacin-SAHA fusion molecule that selectively inhibits COX2 and HDAC6 > HDAC8 > HDAC3 > HDAC2 > HDAC1, inhibits the proliferation of androgen-dependent prostate carcinoma cells [35]. Further studies may find a relevance of HDAC8 inhibition by this compound, because this class I HDAC is also a valid target in AML cells [76]. The finding that compound 11b is effective against cancer cells and selective for COX2 is promising because COX1 is a constitutively expressed housekeeping enzyme while COX2 is induced upon inflammation [35]. Independent thereof is the consensus that the anti-tumor effects of NSAIDs are not directly linked to their inhibitory effects on COX1/COX2, but rather on other effects. These include among others the suppression of β-catenin and aberrant WNT signaling [32,33,35,73].

Caspases -3, -6, -7, and -8 catalyze the degradation of β-catenin [16,17] in solid tumor cells and in vitro. Our data collected with z-VAD-FMK suggest that caspases cleave β-catenin in HDACi-treated leukemic cells. Consistent with this idea, we see β-catenin cleavage products with a similar size as the ones seen by others under apoptotic conditions and caspase activation [16,17]. Further experiments are necessary to decipher which caspases or other enzymes process β-catenin in leukemic cells and why only a subset of HDACi-treated leukemic cells cleaves β-catenin. 

We demonstrate that MYC and WT1 have an antagonistic relationship, with MYC activating WT1 and WT1 suppressing MYC. We also show a regulation of β-catenin through WT1 in leukemic cells. Data collected with kidney-, solid tumor-, and CML-derived cells coherently show that WT1 can repress the expression of MYC [47,48]. However, the opposite effect of WT1 on MYC has also been reported in such cells [54,77]. Culture conditions as well as a context-dependent regulation may explain such discrepancies. Moreover, different isoforms of WT1 can determine whether it has a role as an oncogene or a tumor suppressor [49]. Our data illustrating an accumulation of β-catenin upon a reduction of WT1 in leukemic cells corresponds to an observed negative impact of WT1 on β-catenin signaling in breast cancer cells [51], Sertoli cells forming the blood-testis barrier [50], and podocytes within the kidney [52]. WT1 also impairs β-catenin/WNT signaling via other pathways in vitro and in vivo [53], but one cannot exclude that there is a cell type-specific interplay between β-catenin and WT1 [54]. It likewise remains to be clarified whether the mutual control of MYC and WT1 has further implications, for example for their antagonistic effects on the cell cycle regulator p21 [78,79]. The concomitant loss of all three transcription factors in HDACi-treated cells demonstrates that an inhibition of HDACs, and particularly of HDAC3, overwrites such regulatory effects and leads to their loss by caspase-dependent and -independent mechanisms. Our data additionally suggest that the reduction of MYC by HDACi may lead to a reduction of WT1 in leukemic cells, but not vice versa.

The elimination of β-catenin does not affect MYC and we see no mutual regulation of β-catenin and MYC in leukemic cells. Accordingly, we found that MYC was decreased in HDACi-treated cells irrespective of whether they retained β-catenin. While this lack of coregulation between β-catenin and MYC is unexpected [11,39,40], others also found that an attenuation of β-catenin by BC2059 did not affect MYC levels in leukemic cells [23]. Moreover, this work shows that a similar reduction of β-catenin by LBH589 or BC2059 leads to either a clear or no reduction of MYC. Hence, LBH589 triggers processes that decrease MYC largely independent of β-catenin. It is also possible that the wild-type or mutant status of β-catenin has an impact on its relevance in cells and onto MYC expression. Our data suggest that β-catenin is wild-type in the leukemic cell lines that we analyzed. This finding agrees with the notion that only a subset of AML cases overexpresses β-catenin [4,5,6]. Moreover, an elimination of β-catenin restricts the proliferation of a limited number of AML cells [11] and variable combination indices resulted from LBH589/BC2059 cotreatment schedules in primary AML cells [23]. Furthermore, a dysregulation of β-catenin in leukemic cells does not always imply their addiction to β-catenin [11]. This also holds true for colon cancer cells with mutant β-catenin [56] and β-catenin does not affect a lethal growth of lymphoma in mice [10]. Likewise, if β-catenin was the only critical factor for leukemic cell growth, indomethacin would be far more useful for leukemia treatment.

Our finding that single elimination of β-catenin, MYC, and WT1 does not kill MV4-11 cells may equally hint that HDACi trigger complex apoptosis cascades that cannot be mimicked by a depletion of one stemness-associated transcription factor. Furthermore, a remaining expression of β-catenin, MYC, and WT1 may have ensured the survival of the investigated cells. Maybe a combined inhibition and a more prolonged reduction of these factors halts cell proliferation. Such an idea is supported by the finding that a CRISPR-Cas9-based deletion of β-catenin requires over 10 days to eliminate HEL cells [39]. Since β-catenin cleavage products exert biologically important functions [16], it is equally plausible that RNAi-mediated depletion cannot mimic the effect of HDACi-induced processing of β-catenin in response to a treatment with HDACi. Moreover, other HDACi-triggered mechanisms, including replication stress and DNA damage [21], may be more relevant inducers of cell death. This idea agrees with the finding that HU also triggers the processing of β-catenin.

Remarkably, MYC is of predictive value for therapy success of AML patients who are treated with HDACi [19] and we show that MYC is a target of RGFP966. Whether RGFP966 is useful in the clinic remains to be resolved. In rodents, repetitive applications of RGFP966 were safe and produced beneficial effects without toxicity to normal tissues [65,66,80]. Our data illustrate that a reduction of MYC, irrespective of whether β-catenin is cleaved or not, is a good marker for apoptosis induction by HDACi. In addition to the HDAC1/HDAC2-dependent transcriptional regulation of MYC in pancreatic carcinoma cells [81], our data collected with RGFP966 suggest that HDAC3 supports the expression of MYC in leukemic cells. Consistent with these data, others found that HDAC3 is necessary for the growth of MYC-driven lymphoma [65]. Nonetheless, we cannot rule out that HDAC1 and HDAC2 maintain the expression of β-catenin, MYC, and WT1 in leukemic cells. Moreover, faster migrating products of MYC in LBH589-treated HL60 AML cells suggest a possible proteolysis of MYC [23]. Irrespective thereof, the data that we collected with z-VAD-FMK suggest that HDACi decrease MYC independent of caspase activation.

Regarding WT1, HDACi trigger a loss of the WT1 protein through an induction of the E2 ubiquitin conjugase UBCH8 and a shutdown of the *WT1* promoter [46]. As recent data demonstrate caspase-dependent processing of WT1 upon replication stress and DNA damage in leukemic cells [55,82], caspases may as well degrade WT1 upon an inhibition of class I HDACs in leukemic cells. The details on how HDACi alter the expression, stability, acetylation, and other posttranslational modifications of β-catenin, MYC, and WT1 as well as the physiological consequences thereof should be analyzed in future research.

Taken together, our results show that a loss of β-catenin, MYC, and WT1 are molecular markers for beneficial effects of HDACi. The data also suggest combining class I HDACi with indomethacin against leukemic cells. 

## 4. Materials and Methods

### 4.1. Cell Lines and Reagents

Human leukemic cell lines (MV4-11, HEL, and KG1) were from the German collection of microorganisms and cell cultures. Cells were authenticated by DNA fingerprinting (DNA profiling using eight different and highly polymorphic short tandems repeats) at the Leibniz-Institute (DSMZ, Braunschweig, Germany). Leukemic cells grow at 37 °C and a 5% CO_2_ humidified atmosphere in RPMI 1640 medium supplemented with 5%–10% fetal calf serum (FCS), and 1% penicillin/streptomycin (Sigma-Aldrich, Taufkirchen, Germany). LBH589, MS-275, z-VAD-FMK, and RGFP966 were from Selleckchem (Munich, Germany). Marbostat-100 was synthesized as described [26]. Annexin-V-APC (#550474), 7AAD (#559925) and annexin-V binding buffer (#556454) were provided by BD Bioscience (Heidelberg, Germany). Annexin-V-FITC was from Miltenyi Biotec (Bergisch-Gladbach, Germany), propidium iodide (PI), HU, and indomethacin were from Sigma-Aldrich (Taufkirchen, Germany) and z-VAD-FMK (#FMK001) was provided by R&D Systems (Wiesbaden, Germany). Isolation of mononuclear cells and the subsequent immunomagnetic selection of CD34^+^ cells for establishment of primary AML cell long-term cultures (LTCs) have been described, LTC FFM12 has a monosomy of chromosome 7, and LTC FFM05 has a complex aberrant karyotype [41,83]. AML-LTCs cells were maintained in X-Vivo-10 supplemented with 10% FCS (GE Healthcare HyClone™, München, Germany), interleukin-3, thrombopoietin (25 ng/mL each), stem cell factor, and FLT3-ligand (50 ng/mL each, Miltenyi, Bergisch-Gladbach, Germany) for 7 days. Peripheral blood mononuclear cells (PBMCs) were isolated by Ficoll Histopaque^®^-1077 (Sigma-Aldrich, Taufkirchen, Germany) density gradient centrifugation from buffy coats. Samples were from healthy donors at the blood bank of the University Medical Center Mainz. PBMCs were washed thrice with PBS + 2 mM EDTA + 0.05% BSA and seeded in RPMI 1640 medium containing 10% FCS and 1% penicillin/streptomycin. All cell lines and AML-LTCs were confirmed to be free of mycoplasma every 1–2 months.

### 4.2. Apoptosis and Proliferation Assays 

Since the experiments were conducted in different laboratories, two equally possible methods were used. For the measurement of cell death rates and cell cycle distributions after 24 h treatments, cells were analyzed as described by us [26,29,55]. For the measurements 48 h after treatments, cells were stained using annexin-V-APC (#550474; apoptosis) and 7AAD (#559925; late apoptosis/necrosis) according to the instructions of the manufacturer with annexin-V binding buffer (#556454; BD Bioscience, Heidelberg, Germany) and analyzed by flow cytometry using LSRII or FACS Canto from BD Bioscience (Heidelberg, Germany). For the regrowth assays, 15 × 10^4^ cells/mL were seeded in 24-well plates and treated with RGFP966 or LBH589. Cell numbers and viability of cells were determined at day 4 after discontinuation of treatment by counting and the trypan blue dye exclusion test. Cells were harvested by centrifugation and washed twice with PBS to remove HDACi. 

### 4.3. Western Blotting and Immunofluorescence 

Western blotting was done according to [29,55,84] and the antibodies: Abcam, Berlin, Germany: Anti-HDAC3 (ab16047), anti-WT1 (ab89901), and anti-GAPDH (ab128915); BD Bioscience, Germany: Anti-β-catenin (610153), and anti-poly(ADP)-ribosyltransferase-1 (PARP1) (556362); Cell Signaling Technology, Frankfurt, Germany: Anti-β-actin (#4970), anti-cleaved caspase-3 (9661), anti-MYC (5605), and anti-HDAC6 (7558); Millipore-Merck Darmstadt, Germany: Anti-acetyl-Histone H3 (06-599), anti-HDAC1 (05-100), and anti-γH2AX (05-636); Roche, Mannheim, Germany: Anti-PARP1 (#1835238001); Santa Cruz, Heidelberg, Germany: Anti-β-actin (sc-47778), anti-GSK3β (sc-81462), anti-HDAC2 (sc-7899), anti-HSP90 (sc-13119), anti-p-Ser9-GSK3β (sc-373800), and anti-vinculin (sc-73614); Sigma-Aldrich, Darmstadt, Germany: Anti-acetyl-Tubulin (T7451). Densitometry was performed with Image Studio Lite. 

Immunofluorescence was performed using anti-53BP1 antibody (MAB3802); Millipore-Merck Darmstadt, Germany with the methodology described in [70].

### 4.4. Real-Time PCR-TaqMan (qRT-PCR)

Total RNA and first strand DNA were obtained according to standard protocols. TaqMan-PCR was performed in triplicate using ABI PRISM 7700 (Applied Biosystems, Darmstadt, Germany). “Assays-on-demand” for β-catenin transcripts were used according to the manufacturer’s instructions (Assay-ID: beta-Catenin-Hs01076483_m1; Applied Biosystems, Darmstadt, Germany). Sequence detector software SDS 2.0 (Applied Biosystems, Carlsbad, USA) was used for data analysis. CT values were exported into excel worksheets for calculation of fold changes using the comparative CT method. The internal reference gene was *GUS*, oligonucleotides ENF1102 (*GAAAATATGTGGTTGGAGAGCTCATT*) and ENR1162 (*CCGAGTGAAGATCCCCTTTTTA*), and probe ENPR1142 (Fam-*CCAGCACTCGTCGGT GACTGTTCA*-Tamra).

### 4.5. RNAi

For this genetic manipulation of cells, we used published protocols [29,55]. We used siRNAs at the following concentrations: β-catenin (100 pmol; Cell Signaling Technology, Frankfurt, Germany, #6225), MYC (100 pmol; Santa Cruz, Heidelberg, Germany, sc-29226), and WT1 (100 pmol; Santa Cruz, Heidelberg, Germany, sc-36846). Cells were electroporated with Amaxa™ Nucleofector™ II device using kit L (Lonza, Cologne, Germany, vca-1005) or with the Neon™ Transfection System (Invitrogen, Carlsbad, CA, USA). Cells were analyzed 48 h after electroporation.

### 4.6. Statistics

Statistical analyses were performed with GraphPad Prism 6. Significance was determined with indicated statistical tests. Asterisks are distributed by *p*-values (* *p* ≤ 0.05; ** *p* ≤ 0.01; *** *p* ≤ 0.001; and **** *p* ≤ 0.0001). 

## 5. Conclusions

In conclusion, we demonstrated that low doses of LBH589, MS-275, FK228, and RGFP966 induced apoptosis, replication stress/DNA damage, and a lasting growth arrest of AML cells. Thus, inhibition of HDAC3 seems to be sufficient to compromise the genomic integrity and the survival of such cells. A loss of the transcription factors β-catenin, MYC, and WT1 marks the efficacy of HDACi. HDAC3 activity is a key factor for the maintenance of the expression of these tumor- and stemness-associated transcription factors in AML cells. RNAi-based loss-of-function approaches illustrate a previously unidentified cross-regulation between β-catenin, MYC, and WT1 in leukemic cells. Furthermore, our study suggests that the frequently used non-steroidal anti-inflammatory drug indomethacin potentiates long-lasting anti-proliferative effects of HDACi.

## Figures and Tables

**Figure 1 cancers-11-01436-f001:**
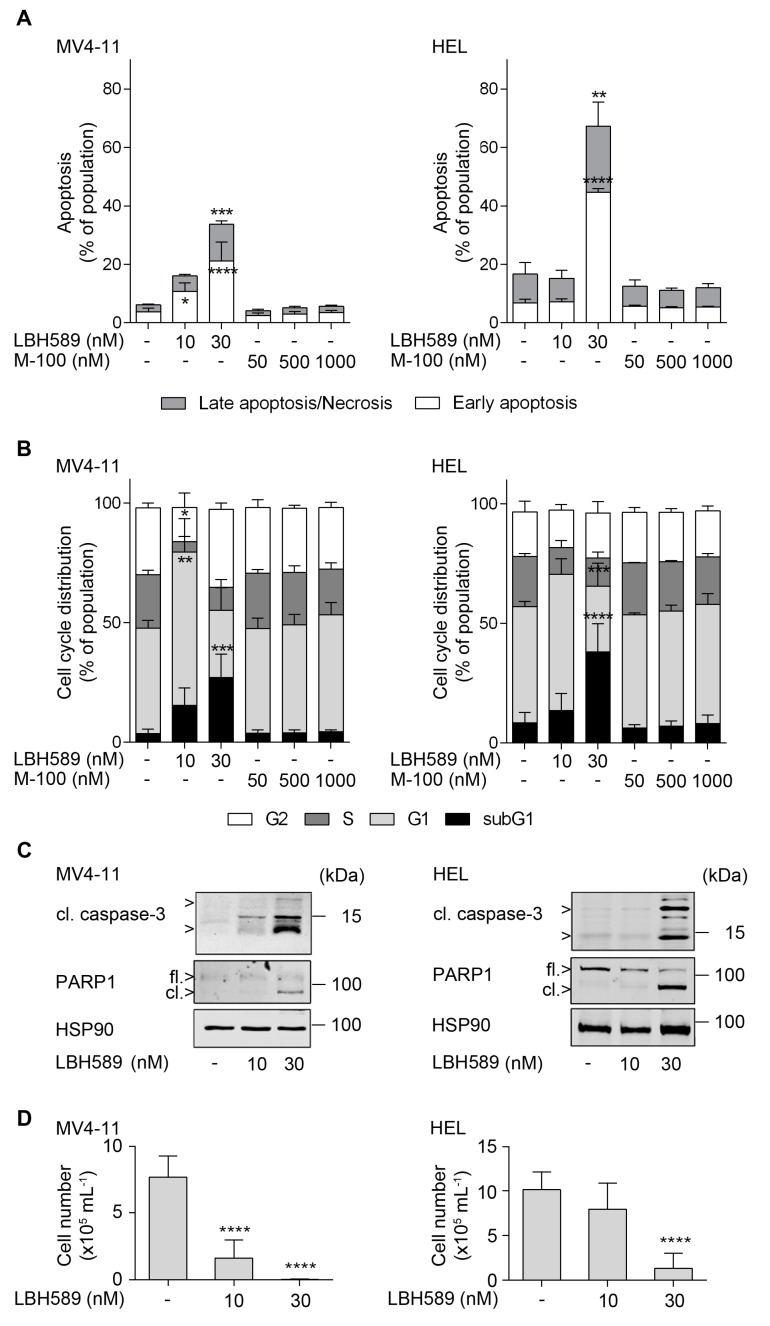
Human leukemia cell lines are sensitive against LBH589. (**A**) Human MV4-11 and HEL cells were treated with 10 to 30 nM LBH589 or increasing doses of marbostat-100 (M-100; 50 nM, 500 nM, and 1 µM) for 24 h. Cells were stained with annexin-V-FITC/propidium iodide (PI) and analyzed by flow cytometry (*n* = 3; mean + SD; two-way ANOVA; * *p* ≤ 0.05; ** *p* ≤ 0.01; *** *p* ≤ 0.001; **** *p* ≤ 0.0001). (**B**) Cells were incubated with HDACi as stated in (**A**). Fixed cells were stained with PI and cell cycle distributions were analyzed by flow cytometry (*n* = 3; mean + SD; two-way ANOVA; * *p* ≤ 0.05; ** *p* ≤ 0.01; *** *p* ≤ 0.001; **** *p* ≤ 0.0001). (**C**) Cells were treated with 10 nM or 30 nM LBH589 for 24 h. Indicated proteins were detected via Western blot (cl., cleaved; fl., full-length) with HSP90 and β-actin as loading controls (*n* = 3). Please note that compared to HEL cells MV4-11 cells have far less full-length PARP1 and that the lot of the anti-PARP1 antibody may have preferentially recognized the cleaved form of PARP1. (**D**) Regrowth of the human leukemia cell lines MV4-11 and HEL. Cells were treated with 10 nM or 30 nM LBH589 for 24 h. Thereafter, cells were washed twice with PBS and reseeded. Cells were stained with trypan blue and viable cells were counted after 4 days (*n* = 3; mean + SD; one-way ANOVA; **** *p* ≤ 0.0001).

**Figure 2 cancers-11-01436-f002:**
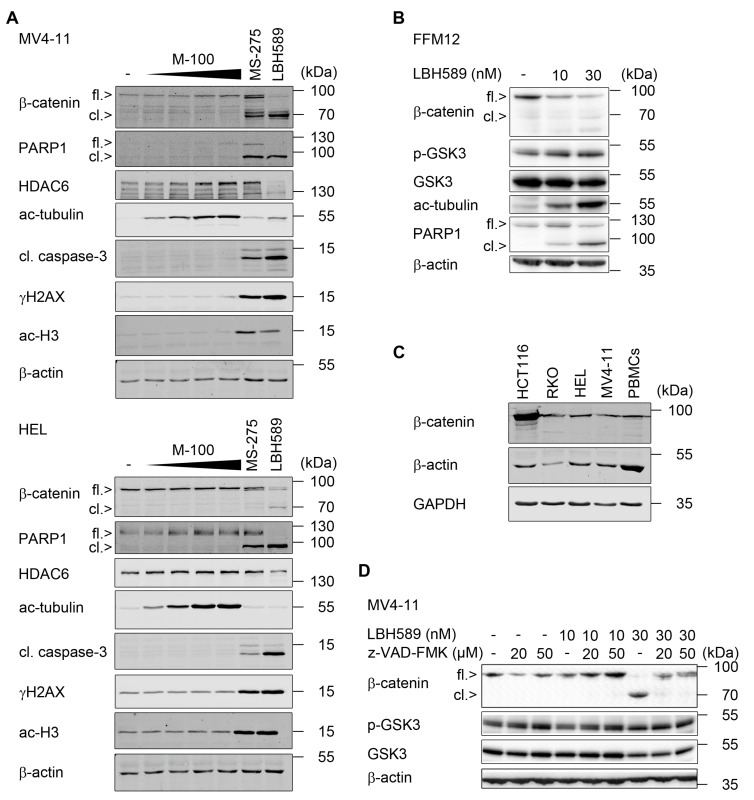
Caspases cleave β-catenin in leukemic cells. (**A**) MV4-11 and HEL cells were treated with increasing doses of marbostat-100 (M-100; 50 nM, 100 nM, 500 nM, and 1 µM), 5 µM of MS-275, or 30 nM LBH589 for 24 h. Whole cell extracts were analyzed by Western blot as indicated (cl., cleaved; ac, acetylated), with β-actin as a loading control (*n* = 2). (**B**) Primary human FFM12 acute myeloid leukemia (AML) cells were incubated with 10 nM or 30 nM LBH589 for 24 h. Whole cell extracts were blotted for β-catenin, phosphorylated (p-) and total GSK3β, and PARP1; β-actin as loading control. (**C**) Indicated human cell lines and peripheral blood mononuclear cells (PBMCs) were compared regarding their β-catenin status. Whole cell extracts were analyzed by Western blot, with GAPDH and β-actin as loading controls (*n* = 2). (**D**) MV4-11 cells were pretreated with 20 µM or 50 µM of the pan-caspase inhibitor z-VAD-FMK for 1 h, followed by stimulation with 10 nM or 30 nM LBH589 for 24 h. Whole cell extracts were blotted for the indicated proteins and β-actin as a loading control (*n* = 2).

**Figure 3 cancers-11-01436-f003:**
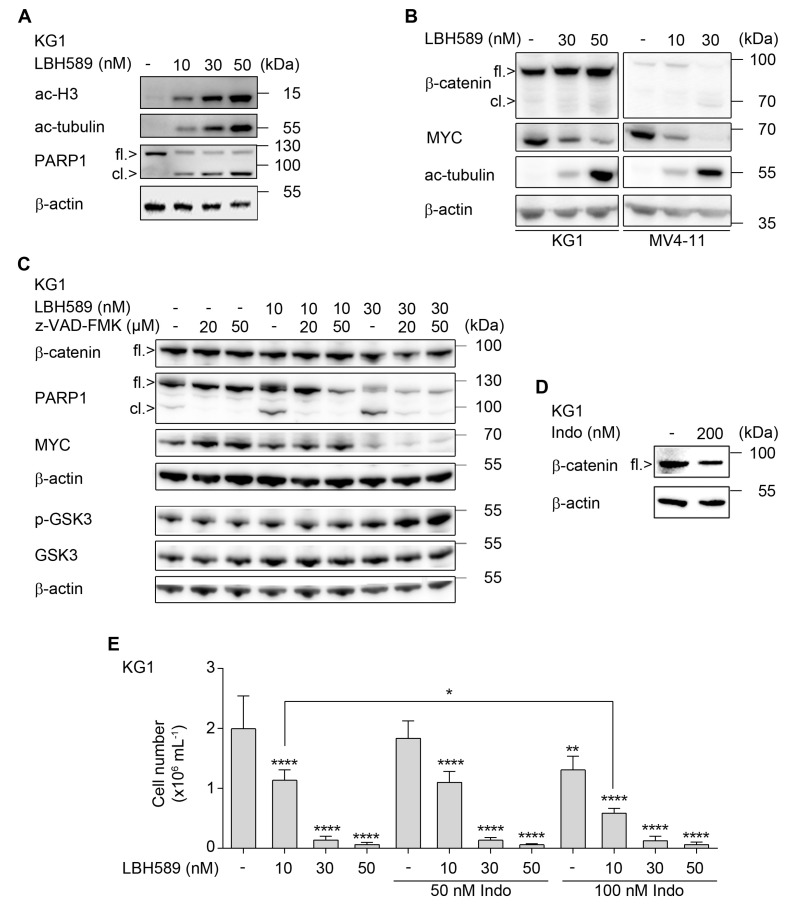
Effects of LBH589 and indomethacin on β-catenin and cell proliferation. (**A**) KG1 cells were incubated with 10 nM, 30 nM, or 50 nM of LBH589 for 24 h. Indicated proteins were detected via Western blot (fl., full-length; cl., cleaved); with β-actin loading as control (*n* = 2). (**B**) KG1 and MV4-11 cells were treated with LBH589 as indicated for 24 h. Western blot was performed for β-catenin, myelocytomatosis oncogene (MYC), acetylated (ac) tubulin, and β-actin as loading control (*n* = 2). (**C**) KG1 cells were incubated with 10–30 nM LBH589 ± 20–50 µM z-VAD-FMK for 24 h. Lysates of these cells were analyzed for β-catenin, full-length/cleaved PARP1 (fl./cl.), and MYC. A separate membrane was incubated with antibodies against total and phosphorylated (p) GSK3β; β-actin was used as loading control (*n* = 2). (**D**) KG1 cells were cultured in the presence or absence of 200 nM indomethacin for 24 h. Expression of β-catenin was assessed by Western blot; with β-actin as loading control (*n* = 2). (**E**) KG1 cells were treated with LBH589 and/or indomethacin as indicated. After 24 h of treatment, substances were washed out and cells were further cultured for proliferation analysis. After 4 days, proliferation was determined by trypan blue exclusion assay. Data are shown as mean values of cells per mL + SD (*n* = 3; one-way ANOVA; * *p* ≤ 0.05; ** *p* ≤ 0.01; **** *p* ≤ 0.0001).

**Figure 4 cancers-11-01436-f004:**
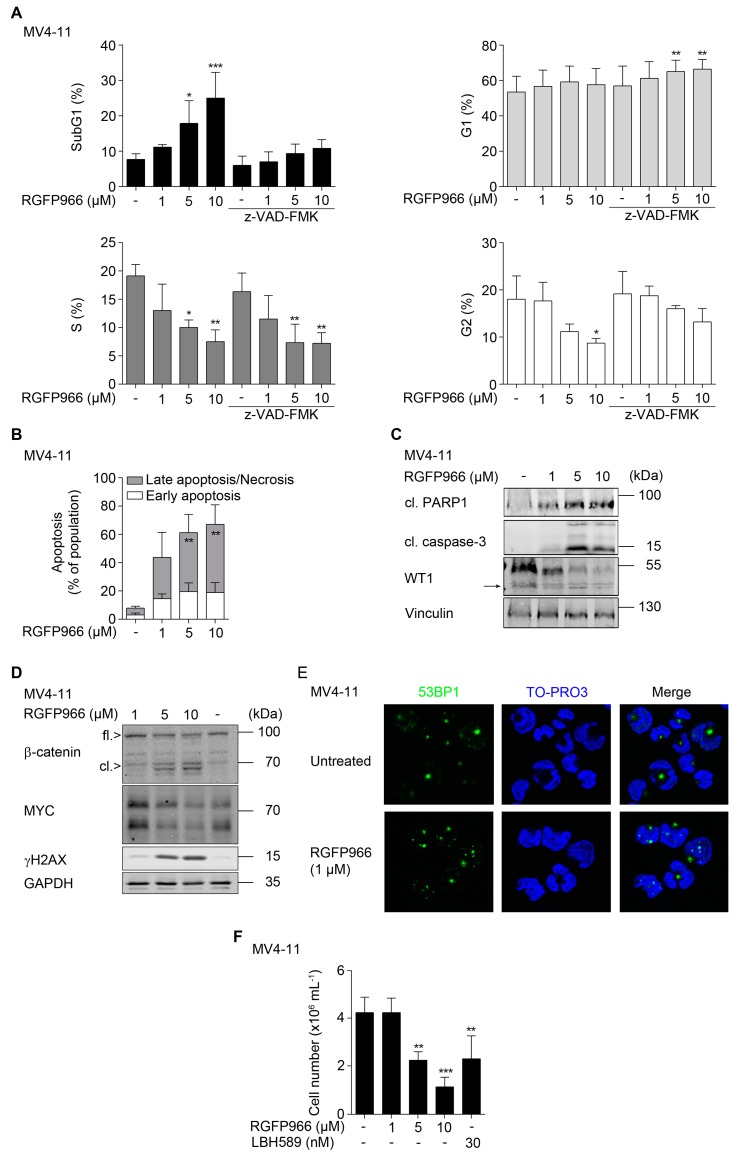
Biological consequences of HDAC3 inhibition and interplay between β-catenin, MYC, and WT1. (**A**) MV4-11 cells were pretreated with z-VAD-FMK for 1 h, followed by incubation with RGFP966 as indicated for 24 h. Cells were fixed and stained with PI and analyzed by flow cytometry (*n* = 3; mean + SD; two-way ANOVA; * *p* ≤ 0.05; ** *p* ≤ 0.01; *** *p* ≤ 0.001). (**B**) MV4-11 cells were treated as indicated for 24 h. After incubation time, cells were stained with annexin-V-FITC/PI and apoptosis was determined by flow cytometry (*n* = 3; mean + SD; two-way ANOVA; ** *p* ≤ 0.01). (**C**) MV4-11 cells were treated with increasing doses of RGFP966 (1 µM, 5 µM, and 10 µM) for 24 h. Caspase-3, PARP1, and WT1 (→ marks WT1 cleavage product) were analyzed by Western blot, with vinculin as loading control (*n* = 2). (**D**) MV4-11 cells were treated as stated in (**C**). Indicated proteins were detected via Western blot, with GAPDH as loading control (*n* = 2). (**E**) Immunofluorescence for 53BP1 in MV4-11 cells that remained untreated or exposed to 1 µM RGFP966 for 24 h. (**F**) MV4-11 cells were treated as indicated for 24 h. Thereafter, cells were harvested and washed with PBS and reseeded. After 4 days, the cell numbers were determined by trypan blue staining and counting (*n* = 3; mean + SD; one-way ANOVA; ** *p* ≤ 0.01; *** *p* ≤ 0.001).

**Figure 5 cancers-11-01436-f005:**
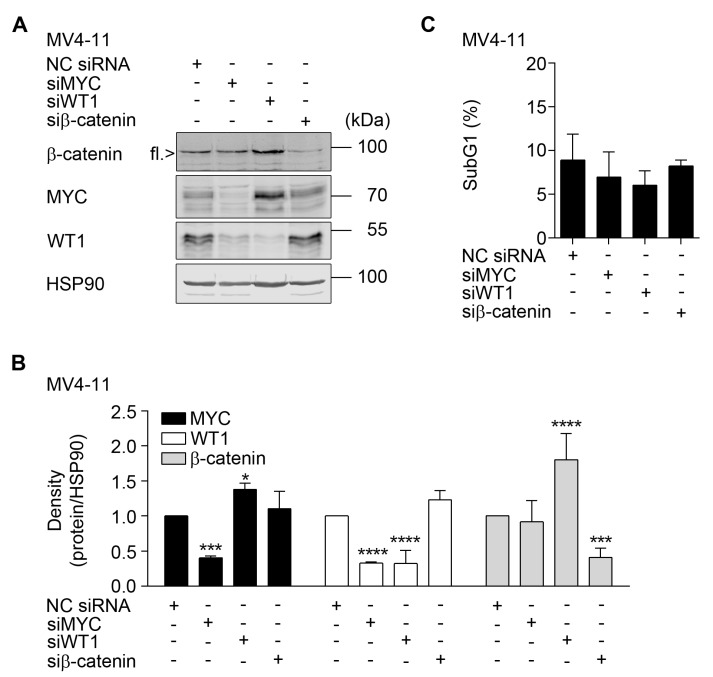
Cross-talk between β-catenin, MYC, and WT1. (**A**) MV4-11 cells were electroporated with 100 pmol of indicated siRNAs and incubated for 48 h (NC siRNA, scrambled irrelevant control siRNA; si, siRNA). Proteins were detected by Western blot, with HSP90 as loading control (*n* = 3). (**B**) Densitometry evaluation of data shown in (A) (*n* = 3; mean + SD; two-way ANOVA; * *p* ≤ 0.05; *** *p* ≤ 0.001; **** *p* ≤ 0.0001). (**C**) Flow cytometry was done with aliquots of MV4-11 cells with an efficient elimination of β-catenin, MYC, or WT1. Cells were fixed, PI-stained, and assessed for apoptotic DNA fragmentation (*n* = 3; two-way ANOVA; not significant).

## Data Availability

All relevant data are within the paper.

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
