# Peer review of "HDAC3 Activity is Essential for Human Leukemic Cell Growth and the Expression of β-catenin, MYC, and WT1"

_cancers, 2019, doi:10.3390/cancers11101436_

Round 1

Reviewer 1 Report

The manuscript entitled “HDAC3 is Essential for Human Leukemic Cell Growth and the Expression of β-catenin, MYC, and WT1” by Beyer et al. is clearly written, very well structured and of high interest to researchers of the HDAC field. The authors investigated, how different HDAC inhibitors, covering different HDAC isozymes (from pan to isotype-selective), affect the proliferation and survival of established and primary leukemic cells. The authors show that HDACi against class I HDACs and especially the HDAC3 selective compound RGFP966 suppress tumor cell proliferation and induce apoptosis as well as DNA stress in these cell models. They demonstrate the modulation of expression of important leukemogenic transcription factors, such as beta-catenin, c-MYC, and WT1. Moreover, the authors reveal a very favorable combination of the anti-inflammatory drug indomethacin with HDACi against leukemic cells. The data are convincing and well presented. I have just some minor suggestions:

The HDAC class I family comprises HDAC1, 2, 3 and HDAC8. The authors should discuss the role of HDAC8 for the survival of leukemic cells. Pan-HDAC inhibitors, such as Panobinostat can - in high concentrations - be toxic to non-tumor cells as well. One benefit of the usage of more selective inhibitors is the reduction in toxicity. Are the RGFP966 doses that effectively kill leukemic cells non-toxic to normal human cells? Figure 3: it seems that panel 3F is missing. The description of 3F is in the Figure legend, but no panel F is visible in the Figure. Figure 4 F, G: It is not clear what “siRNA-B” means – is it the negative control siRNA? I suggest changing the label of this sample to “NC siRNA”. What was the viability of the leukemic cells upon transfection with siRNAs? For the discussion: what is known so far about the specific role of HDAC3 and the genetic depletion of HDAC3 in the context of leukemic cells?

Author Response

Reviewer 1

Our response: We would like to thank all four reviewers for their thorough evaluation of our manuscript and their constructive comments. We provide point-by-point responses to all issues raised.

The HDAC class I family comprises HDAC1, 2, 3 and HDAC8. The authors should discuss the role of HDAC8 for the survival of leukemic cells.

Our response: We thank the reviewer for pointing out this oversight to us. Our revised manuscript now contains a discussion on HDAC8. We discuss that HDAC8 is a valid target in AML cells (Qi et al., Cell Stem Cell. 2015) and that a co-inhibition of HDAC8, HDAC6, and HDAC10 can induce DNA damage and suppress tumor growth (Kolbinger et al., Archives of Toxicology 2018). Moreover, we cite recent reviews that sum up how specific inhibition of HDACs can be beneficial (Koeneke et al., Cells 2015; Cappellacci et al., Current medicinal chemistry 2018).

Are the RGFP966 doses that effectively kill leukemic cells non-toxic to normal human cells?

Our response: RGFP966 is an experimental drug that has not been tested in clinical trials. However, RGFP966 has been applied to mice without generating significant toxicity. For example, Schmitt et al., J Ocul Pharmacol Ther. 2018 state that: “…Repeated IP injections of 2 mg/kg RGFP966 over the course of 2 and 4 weeks post ONC prevented RGC loss. There were no significant toxic or antiproliferative effects to off-target tissues in mice treated daily for 14 days with RGFP966”. Matthews et al., Blood 2015 write: “Treatment of wild-type hematopoietic progenitor cells with RGFP966 had only minor effects on cell clonogenicity”; this paper also shows the efficacy of RGFP966 against transformed blood cells.

We agree that such toxicological considerations are important. Therefore, we added the above-mentioned publications and discuss whether RGFP966 can be applied safely in vivo.

Figure 3: it seems that panel 3F is missing. The description of 3F is in the Figure legend, but no panel F is visible in the Figure.

Our response: We are really sorry for the mistake, this was from a previous manuscript version. We have corrected the figure.

Figure 4 F, G: It is not clear what “siRNA-B” means – is it the negative control siRNA? I suggest changing the label of this sample to “NC siRNA”.

Our response: We agree and changed the labeling as suggested.

What was the viability of the leukemic cells upon transfection with siRNAs?

Our response: We now present these data, which we had mentioned as (data not shown) in the previous version of the manuscript, as new Fig. 5C.

For the discussion: what is known so far about the specific role of HDAC3 and the genetic depletion of HDAC3 in the context of leukemic cells?

Our response: A recent review summarizes functions of HDAC3 that are dependent and independent of its catalytic activity (Emmett and Lazar, Nature Rev. Mol. Cell. Biol. 2019). For our work it is important that RGFP966 can phenocopy a genetic depletion of HDAC3 (Matthews et al., Blood 2015). Moreover, we applied RGFP966 at a maximal dose of 10 µM, because it does not inhibit any other HDAC at a dose up to 15 µM (Malvaez et al., PNAS 2012). We discuss the point raised by the referee in our revised manuscript. Moreover, we have realized that we should stress that our manuscript demonstrate a pharmacologically feasible way to interfere with leukemic cell growth. Therefore, we added “activity” to the title: HDAC3 Activity is essential for Human Leukemic Cell Growth and the Expression of β-catenin, MYC, and WT1.

Reviewer 2 Report

In the paper HDAC3 is essential for human leukemic cell growth and the expression of β-catenin, MYC and WT1 Beyer and co-workers aimed to elucidate the therapeutic potential of inhibitors of histone deacetylases using a number of permanent human leukemic cell lines as well as primary cultures of acute myeloid leukemia (AML). Furthermore, they investigated the relationship between β-catenin, MYC and WT1 and how a specific HDAC3 inhibitor affects their regulation. The authors report that reduced levels of β-catenin, MYC and WT1 are molecular markers for the efficacy of HDACi. They claim that targeting specifically HDAC3 might be beneficial to reduce cell survival of AML cells. The experiments are well performed and the topic is highly relevant. However, there are a number of points the authors should address before considering the manuscript for publication in “Cancers”.

Major points:

Figure 3F: The Figure is missing in the manuscript.

Figure 4F: It is difficult to judge from the presented immunoblot to what extent siRNA against β-catenin reduces the level of β-catenin. It is therefore not possible to answer the question whether β-catenin affects MYC and WT1 and to claim that β-catenin does not modulate MYC (Page 11, line 301). The authors themselves state that there might be “insufficient reduction by RNAi” (line 375). In general, to make any judgement about an interaction between β-catenin, MYC and WT1, I would suggest quantification of the si-mediated downregulation of β-catenin, MYC and WT1 observed on the two immunoblots which were performed (according to figure 4 legend). The quantification data should be included in the manuscript. Based on the data regarding the levels of β-catenin, MYC and WT1 the authors might be able to draw a conclusion regarding an interplay between β-catenin, MYC and WT1.

Discussion, line 349: The authors state that MYC activates WT1 while WT1 suppresses MYC (according to Figure 4G). They also state that the reduction of MYC by HDACi may lead to a reduction of WT1 in leukemic cells but not vice versa (line 350). However, their own data show that HDACi leads to reduced levels of WT1 and MYC (Figure 4 D,E). This is rather confusing. If WT1 really suppresses MYC, how do they explain that mediated by HDACi both proteins are present at reduced levels? Is the effect HDACi has on proteins superior to the antagonistic relationship between MYC and WT1? If there is any hypothesis I would suggest the authors add it to the discussion section.

Minor points:

Introduction, page 2, line 45: The authors wrote that “21.000 people will be diagnosed with this disease”. Having read the abstract, the reader would of course know that the disease in question is acute myeloid leukemia (AML), but this has not been mentioned in the Introduction section so far. I would recommend to the authors to include the information at that point.

Figure 4 F, G: “siRNA-B” is not mentioned in the legend. I guess it shows the bands for the respective control siRNAs but this should be included in the figure legend.

Supplemental Figure 1: instead of showing an increasing bar for treatment with LBH589 I would suggest that the authors include the exact dose, as presented in Figure 1A.

Supplemental Figure 3: the figure is not mentioned in the manuscript.

Supplement Figure 2: a strong signal for γH2AX can be observed in MV4-11 cells treated with 10 nM FK228. This signal is abrogated when cells were pretreated with the caspase inhibitor z-VAD-FMK. However, as γH2AX is an indication for DNA damage it is difficult to understand why a caspase inhibitor would inhibit phosphorylation of γH2AX. Do the authors have any explanation for this observation? As the authors stated “this is not merely a consequence of apoptosis” (line 322) it might be related to other cellular factors, such as replication stress. The authors should briefly explain the observation.

Author Response

Reviewer 2

Our response: We would like to thank all four reviewers for their thorough evaluation of our manuscript and their constructive comments. We provide point-by-point responses to all issues raised.

Major points:

Figure 3F: The Figure is missing in the manuscript.

Our response: We are really sorry for the mistake, this was from a previous manuscript version. We have corrected the figure.

Figure 4F: It is difficult to judge from the presented immunoblot to what extent siRNA against β-catenin reduces the level of β-catenin. It is therefore not possible to answer the question whether β-catenin affects MYC and WT1 and to claim that β-catenin does not modulate MYC (Page 11, line 301). The authors themselves state that there might be “insufficient reduction by RNAi” (line 375). In general, to make any judgement about an interaction between β-catenin, MYC and WT1, I would suggest quantification of the si-mediated downregulation of β-catenin, MYC and WT1 observed on the two immunoblots which were performed (according to figure 4 legend). The quantification data should be included in the manuscript. Based on the data regarding the levels of β-catenin, MYC and WT1 the authors might be able to draw a conclusion regarding an interplay between β-catenin, MYC and WT1.

Our response: We agree that this point needs careful evaluation and discussion. We quantified the downregulation of the transcription factors by the siRNAs. To do this adequately, we repeated the experiment. We used a more efficient protocol that we had established in the last weeks with a new machine (Neon™ Transfection System, Invitrogen, Carlsbad, California, USA). We could fully confirm our results and present them together with the statistical evaluation of the Western blot bands (n=3), see new Fig. 5A-B. We could suppress the levels of all three transcription factors about equally well and to a significant extent (Fig. 5B). Moreover, we discuss how our data can be integrated into the current knowledge on the interplay between β-catenin, MYC, and WT1.

Discussion, line 349: The authors state that MYC activates WT1 while WT1 suppresses MYC (according to Figure 4G). They also state that the reduction of MYC by HDACi may lead to a reduction of WT1 in leukemic cells but not vice versa (line 350). However, their own data show that HDACi leads to reduced levels of WT1 and MYC (Figure 4 D,E). This is rather confusing. If WT1 really suppresses MYC, how do they explain that mediated by HDACi both proteins are present at reduced levels? Is the effect HDACi has on proteins superior to the antagonistic relationship between MYC and WT1? If there is any hypothesis I would suggest the authors add it to the discussion section.

Our response: Indeed, this is an interesting point and we thank the reviewer for pointing this out to us. The different outcomes between RNAi against MYC and WT1 and the treatment with pan-HDACi, class I HDACi, and the HDAC3 inhibitor can be explained by different conditions. In the RNAi setting, the cells are resting. However, in the HDACi treatment schedules, the cells have altered acetylation profiles and consequently altered signaling and gene expression. These lead to a caspase-dependent processing of β-catenin and WT1 and a caspase-independent mechanism that decreases MYC. We discuss this in the revised manuscript.

Minor points:

Introduction, page 2, line 45: The authors wrote that “21.000 people will be diagnosed with this disease”. Having read the abstract, the reader would of course know that the disease in question is acute myeloid leukemia (AML), but this has not been mentioned in the Introduction section so far. I would recommend to the authors to include the information at that point.

Our response: Indeed, this is a good point, we have corrected the sentence to “…It is estimated that there will be 437,033 new cases of leukemia and 309,006 deaths associated worldwide in 2018 [1]. Predictions state that more than 21,450 people will be diagnosed with AML and nearly 10,920 people will die from it in alone in the USA in 2019 [2]…”. We believe that we increases the impact of our work by mentioning the worldwide number of people affected by AML.

Figure 4 F, G: “siRNA-B” is not mentioned in the legend. I guess it shows the bands for the respective control siRNAs but this should be included in the figure legend.

Our response: We agree that this should be clarified. We changed the labeling (as also suggested by Reviewer 1 to NC siRNA) and explain this abbreviation.

Supplemental Figure 1: instead of showing an increasing bar for treatment with LBH589 I would suggest that the authors include the exact dose, as presented in Figure 1A.

Our response: We changed this as requested.

Supplemental Figure 3: the figure is not mentioned in the manuscript.

Our response: We are thankful for pointing out this flaw to us and now mention it the manuscript.

Supplement Figure 2: a strong signal for γH2AX can be observed in MV4-11 cells treated with 10 nM FK228. This signal is abrogated when cells were pretreated with the caspase inhibitor z-VAD-FMK. However, as γH2AX is an indication for DNA damage it is difficult to understand why a caspase inhibitor would inhibit phosphorylation of γH2AX. Do the authors have any explanation for this observation? As the authors stated “this is not merely a consequence of apoptosis” (line 322) it might be related to other cellular factors, such as replication stress. The authors should briefly explain the observation.

Our response: This is a very critical issue. We have to admit that we made a mistake in the labeling of the figure. The z-VAD-FMK lane is the last one and this is now corrected:

Looking at the corrected figure, it now becomes evident that z-VAD-FMK can hardly prevent the accumulation of ɣH2AX. As this concentration of z-VAD-FMK though abrogates the cleavage of β-catenin, it is effective. Therefore, we conclude that this accumulation of ɣH2AX is not prevented when caspase activation is blocked. We addressed this, because the multitude of DSBs generated upon DNA fragmentation by caspase-activated DNase during apoptosis can also induce ɣH2AX (Huang et al., Cell Prolif. 2005; this pathway was originally identified by Rogakou et al., J. Biological Chemistry 1998). We clarify this in the revised manuscript, including a citation of these articles.

Moreover, to assess if the increase in ɣH2AX is linked to DNA damage or a consequence of DNA fragmentation by apoptosis, we incubated MV4-11 cells with 1 µM RGFP966 as a dose that does not evoke significant DNA damage and stained cells for 53BP1 foci by immunofluorescence. We chose this marker by virtue of its ability to stain DNA breaks within foci. We could readily detect 53BP1 foci in RGFP966-treated MV4-11 cells, indicating that HDACi-induced DNA damage is not just a consequence of apoptotic DNA cleavage. These data are included in the revised manuscript, see Fig. 4A,E.

Reviewer 3 Report

The manuscript titled, "HDAC3 is essential for Human Leukemic Cell Growth and the Expression of β-catenin, MYC, and WT1" authored Beyer et al., is an interesting work. The problem statement and the aim of the project were established and the work highlights the importance in identifying the markers to identify the efficacy of HDAC inhibitors.

Concerns.

1. Authors must repeat some of the westerns especially, the cleaved caspase-3 blot in Figure 1C, is not properly cropped and in the same figure, in control there is no full PARP at all, if there is no PARP signal in control, how can we validate that LBH589 cleaved it.  Same thing in various figures, PARP signal in Figure 2A, for both MV4-11 and HEL. cleaved caspase-3 and cl-PARP in Figure 4E

2. The apoptosis data is presented differently in Figure 3 unlike in rest of the figures. 

3. the apoptosis data in Figure 4B, indicates there is no much change in early apoptotic cells. Late apoptottic cells could be from necrosis as well, so it could be necrotic cell death as well . I recommend reporting % cells in 7AAD positive and APC negative quadrant as well. Or I highly recommend presenting some (if not all) dot plots of apoptosis data.

4. Formatting, grammar and spell check is suggested. Including the title there are extra spaces at several places. A thorough spell check si recommended along with grammar check.

Author Response

Reviewer 3

Our response: We would like to thank all four reviewers for their thorough evaluation of our manuscript and their constructive comments. We provide point-by-point responses to all issues raised.

Concerns

Authors must repeat some of the westerns especially, the cleaved caspase-3 blot in Figure 1C, is not properly cropped and in the same figure, in control there is no full PARP at all, if there is no PARP signal in control, how can we validate that LBH589 cleaved it.  Same thing in various figures, PARP signal in Figure 2A, for both MV4-11 and HEL. cleaved caspase-3 and cl-PARP in Figure 4E.

Our response:

Caspase-3:

We have cropped the figure according to the reviewer’s recommendation:

We would like to point out that our figure presenting several caspase-3 cleavage fragments is consistent with the literature. One of the first reports on caspase-3 activation (caspase-3 called CPP32 at that time) is Fernandes-Alnemri et al., PNAS 1996. They report that caspase-3 is processed for activation into p20, p19, p17, and p12 (p17 not specifically annotated in the left panel, but mentioned in the right panel):

Companies offering caspase-3 antibodies also show several bands, please see the following example:

PARP1:

We agree that this requires an explanation. The PARP1 antibody that we used for these blots (BD Biosciences 556362) recognizes full-length and cleaved PARP1. We assume that it recognizes the cleaved form better than the full-length form. Alternatively, PARP1 might be induced before its cleavage in HDACi-treated cells. Both can explain why the signal for cleaved PARP1 exceeds the signal for the full-length protein. We have incorporated this discussion to the legend to the figure. Clarifying the antibody specificity or the regulation of PARP1 is beyond the scope of our manuscript, but we hope that we have answered the referee’s question. Moreover, we added a Western blot result in which we compared MV4-11 cells and HEL cells side-by-side (50 µg per lane). As expected from the individual analyses, HEL cells express much higher levels of PARP1. This can explain the different levels seen in the separate analyses:

The apoptosis data is presented differently in Figure 3 unlike in rest of the figures. 

Our response: The flow cytometry data shown in Figure 3 were collected in the laboratory of Dr. Bug by Dr. Romanski. The cells were stained using annexin-V-APC (#550474; apoptosis) and 7AAD (#559925; late apoptosis/necrosis) according to the instructions of the manufacturer with annexin-V binding buffer (#556454; BD Bioscience, Germany), but no gating was done. All other flow cytometry analyses were done with annexin-V/PI in my group by M. Beyer, A. Mustafa, and M. Pons. Therefore, we repeated the experiment as a technical triplicate (Supplemental Figure 4). We realized that this data contains basically the same information as the previous Fig. 3B. Therefore, we assembled the figure in a way that the former Fig. 3B is now Fig. 3A and the flow cytometry is now presented as Supplemental Figure 4.

The apoptosis data in Figure 4B, indicates there is no much change in early apoptotic cells. Late apoptottic cells could be from necrosis as well, so it could be necrotic cell death as well. I recommend reporting % cells in 7AAD positive and APC negative quadrant as well. Or I highly recommend presenting some (if not all) dot plots of apoptosis data.

Our response: We entirely agree with the reviewer. Therefore, we now present examples of original dot blots for all apoptosis measurements as Supplementary information.

Formatting, grammar and spell check is suggested. Including the title there are extra spaces at several places. A thorough spell check si recommended along with grammar check.

Our response: We found only three extra spaces and removed them. Perhaps, other extra spaces were introduced by to the conversion from the .docx to the .pdf. We have carefully checked the manuscript for grammar and spelling mistakes and we have thoroughly revised it; please see manuscript version with the changes marked up.

Reviewer 4 Report

Page 2, line 71: Please make clear here again that LBH589 is panobinostat which was mentioned two sentences before.

Discussion: Please cite and discuss the HDAC-inhibiting indomethacin conjugates disclosed by Oyelere here (Bioorg. Med. Chem. 2017, 25, 1202-1218).

The manuscript is well written and well designed. The experiments are accurate, the results are well presented, the figures are clear and the conclusions are reasonable. That's why I recommend acceptance of the manuscript after minor revision.

Author Response

Reviewer 4

Our response: We would like to thank all four reviewers for their thorough evaluation of our manuscript and their constructive comments. We provide point-by-point responses to all issues raised.

Comments and Suggestions for Authors

Page 2, line 71: Please make clear here again that LBH589 is panobinostat which was mentioned two sentences before.

Our response: We amended this as suggested, we used abbreviations for LBH589 and FK228 as these are explained in the abstract and we introduced new abbreviations as necessary (“…four HDACi (LBH589, FK228, vorinostat [SAHA], and belinostat [PXD101]) have been approved by the FDA…”).

Discussion: Please cite and discuss the HDAC-inhibiting indomethacin conjugates disclosed by Oyelere here (Bioorg. Med. Chem. 2017, 25, 1202-1218).

Our response: We fully agree that this is a very important publication. Moreover, it is nice that the data that Raji…Oyelere collected in various solid tumor cells support the findings that we made in leukemic cells. We cite this work and discuss its implications in detail. We have additionally incorporated the works by Li et al., Molecular Cancer Therapeutics 2013 and Wang et al., Molecular carcinogenesis 2011 (which are cited by Raji… Oyelere in Bioorg. Med. Chem. 2017, 25, 1202-1218) in our revised manuscript.

Round 2

Reviewer 3 Report

I appreciate the authors for their prompt reply to all the concerns. The responses are satisfactory.